# Development of Engineered Cementitious Composites (ECCs) Incorporating Iron Ore Tailings as Eco-Friendly Aggregates

Kangning Liu [1], Sheliang Wang [1,*], Xiaoyi Quan [1], Jing Wu [1], Jin Xu [1], Nan Zhao [1] and Bo Liu [2]

1   School of Civil Engineering, Xi'an University of Architecture & Technology, Xi'an 710055, China; lkn@xauat.edu.cn (K.L.); quanxy@xauat.edu.cn (X.Q.); wj@xauat.edu.cn (J.W.); 20160030@xijing.edu.cn (J.X.); zn@xauat.edu.cn (N.Z.)
2   China Railway 20th Bureau Group Co., Ltd., Xi'an 710016, China; liubo208@xauat.edu.cn
*   Correspondence: sheliangwang@xauat.edu.cn

**Abstract:** In this study, iron ore tailings (IOTs) are used as aggregates to prepare iron-ore-tailing-engineered cementitious composites (IOT-ECCs) to achieve clean production. Some mechanical indexes, such as compressive strength ($f_{cu}$), four-point flexural strength ($f_f$), axial compressive strength ($f_c$), deformation properties, flexural toughness, and stress–strain behavior, are studied. The mass loss, $f_{cu}$ loss, relative dynamic modulus elasticity (RDEM), and deterioration mechanism after the sulfate freeze–thaw (F-T) cycle are discussed in detail. In addition, pore structure analysis is performed using nuclear magnetic resonance (NMR), while a scanning electron microscope (SEM) is utilized to study the micro-morphology. The results showed that under the 20–80% IOT replacement ratio, IOT-ECCs exhibited improvements in their mechanical properties, pore structure, and resistance to sulfate freeze–thaw (F-T). The most notable mechanical properties and sulfate F-T resistance were demonstrated by the IOT-ECC with 40% IOTs (namely, IOT40-P2.0). Meanwhile, IOT40-P2.0 exhibited good pore structure as well as the bonding interface of the PF and the matrix. The pore structure and compactness of the matrix of IOT-ECCs gradually deteriorated as the F-T cycle increased. The research results will promote the application of IOTs in ECCs.

**Keywords:** iron ore tailings; engineered cementitious composites; mechanical properties; sulfate freeze–thaw cycle resistance; pore structure; microscope characteristics

## 1. Introduction

Engineered cementitious composites (ECCs), a type of cement-based composite material, demonstrate exceptional tensile ductility [1–3]. It was reported that the deformability of ECCs with 2% polyvinyl alcohol (PVA) fiber (PF) can reach 3–5% [3]. The ability of outstanding tensile ductility and multi-crack expansion result in significant enhancements to their mechanical properties and durability, compared to normal concrete. However, the raw material silica sand (S) used to prepare ECCs is relatively expensive, which leads to an increase in the production cost of ECCs and restricts their wider application. In addition, the manufacture of S emits a large amount of $CO_2$, which creates a greater environmental burden. Therefore, looking for a replacement for S to produce eco-friendly ECCs has become a popular research topic [4,5]. To date, some waste materials, including waste glass aggregate [6], recycled fine aggregates [7], limestone powder [8], etc., have been used to produce green ECCs, and it has been found that industrial waste as an eco-friendly substitute for the preparation of ECCs can not only reduce the excessive consumption of natural resources but also reduces the production cost of ECCs.

Iron ore tailings (IOTs), as one of the main industrial by-products, are gradually growing, which results in both land occupation and soil pollution, seriously affecting the ecological environment [9]. The effective utilization of IOTs has aroused great concern globally [10,11]. At present, some studies have found that because the composition is

similar to that of S and the microstructure and quality are relatively stable, IOTs have better potential to be used as an alternative to S in preparing IOT cementitious composites than other industrial by-products [12]. Meanwhile, some researchers have found that adding IOTs can improve the performance of the cementitious composite [13,14]. For example, Shettima et al. [15] found that some strength indexes of concrete mixed with IOTs are better than those of the control group (without IOTs), but the workability is the opposite. Jz et al. [16] reported that the workability, mechanical properties, and durability of concrete are improved by adding appropriate replacement IOTs. Similarly, Liu et al. [17] found that the ideal IOT replacement ratio is 40% in concrete. The concrete incorporating 40% IOTs had the lowest porosity and a denser microstructure, which exhibited the best mechanical properties and sulfate attack, and dry–wet cycle resistance. In addition, Huang et al. [18] prepared ECCs by incorporating IOTs with different particle sizes as aggregates and reported that when the particle size of the IOTs used was within a reasonable range, the same mechanical properties as conventional ECCs could be obtained. Previous studies have found that industrial by-product IOTs have a good prospect for the production of cementitious composites as potential aggregates.

However, previous studies found that there are still few studies on using IOTs as a fine aggregate to prepare ECCs. It is noteworthy that limited studies on iron-ore-tailing-engineered cementitious composites (IOT-ECCs) have focused on the effects of IOTs on the macroscopic mechanical properties of ECCs, while studies on the effects of sulfate freeze–thaw resistance, pore structure from nuclear magnetic resonance (NMR), and microstructure are relatively scarce. It is necessary to conduct more in-depth research on IOT-ECC from mesoscopic and microscopic points of view. This paper provides a comprehensive evaluation of IOT-ECCs prepared by replacing S with IOTs. It mainly explains the influence of IOTs on macroscopic mechanical properties and sulfate freeze–thaw resistance through the analysis method of the combination of pore structure and microstructure. The results of this study will broaden the existing literature on IOTs and promote the application of IOTs in ECCs.

Thus, this paper is devoted to carrying out comprehensive research on IOT-ECCs from both mesoscopic and microscopic points of view. Firstly, the main components, micro-morphology, and physical properties of the IOTs were tested. Secondly, the mechanical properties including the compressive strength ($f_{cu}$), four-point flexural strength ($f_f$), axial compressive strength ($f_c$), deformation properties, flexural toughness, and stress–strain behavior of IOT-ECCs were analyzed in detail. At the same time, the test of sulfate F-T cycles was carried out. Some indexes including mass loss, $f_{cu}$ loss, and relative dynamic elastic modulus (RDEM) of IOT-ECCs were measured. Finally, nuclear magnetic resonance (NMR) was employed to analyze pore structure, and the scanning electron microscope (SEM) was utilized to conduct microscopic characteristics analysis.

## 2. Materials and Methods

### 2.1. Materials

2.1.1. Cementitious Material

The cementitious materials included cement (PO42.5R of Type II), metakaolin, and first-grade fly ash. The chemical composition of the three materials are shown in Table 1.

**Table 1.** Chemical compositions of cement, metakaolin, and fly ash.

| | Chemical Compositions | | | | | | | |
|---|---|---|---|---|---|---|---|---|
| | $SiO_2$ | $Al_2O_3$ | CaO | MgO | $Fe_2O_3$ | $TiO_2$ | $SO_3$ | Others |
| Cement | 55.7% | 42.5% | 0.4% | | 0.3% | 0.9% | | 0.2% |
| Metakaolin | 22.1% | 5.0% | 63.8% | 0.9% | 5.5% | | 2.1% | 0.6% |
| Fly ash | 52.97% | 29.96% | 3.66% | 1.52% | 7.98% | | 0.65% | 3.26% |

### 2.1.2. Fine Aggregate

The fine aggregate used in this study comprised S and IOTs obtained from the Yaogou tailings pond in Shaanxi Province, and the main physical characteristics of S and IOTs are listed in Table 2. Meanwhile, the XRD and SEM techniques were employed to determine the main components and micro-morphology of S and IOTs, as shown in Figure 1. Figure 2 presents the particle size distribution curves of S and IOTs.

**Table 2.** Physical properties of S and IOTs.

| Aggregates | Apparent Density (kg m$^{-3}$) | Loose Bulk Density (kg m$^{-3}$) | Crushing Index (%) | Water Absorption (%) | Mud Content (%) | Water Content (%) |
|---|---|---|---|---|---|---|
| S | 2756 | 1844 | 16.45 | 2.16 | 0.5 | 1.48 |
| IOTs | 2734 | 1827 | 20.37 | 8.9 | 2.6 | 4.36 |

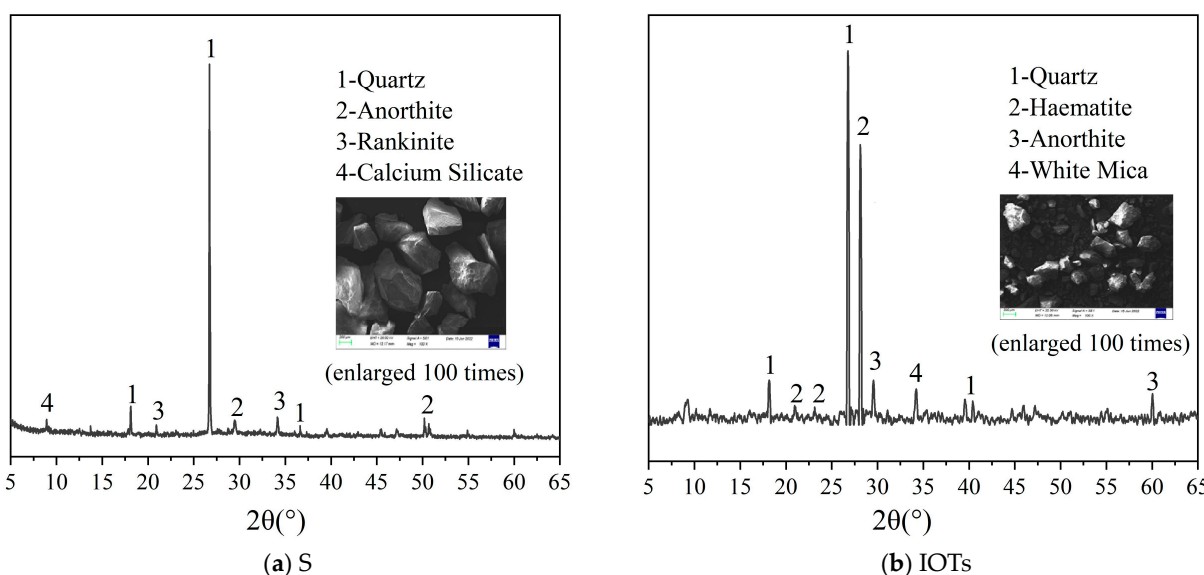

(**a**) S  (**b**) IOTs

**Figure 1.** Microscopic analysis of S and IOTs.

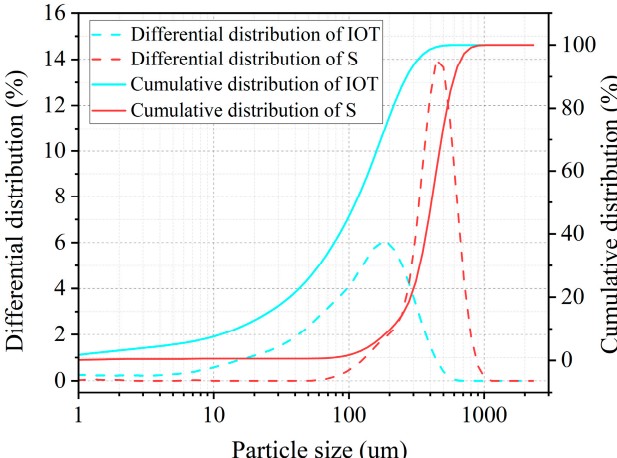

**Figure 2.** Aggregate particle size curves.

2.1.3. Fiber

The fiber used was PF, with its characteristic parameters shown in Table 3.

**Table 3.** Characteristic parameters of fiber.

| Materials | Length (mm) | Diameter (mm) | Density (g cm$^{-3}$) | Fracture Elongation (%) | Tensile Strength (MPa) | Modulus of Elasticity (GPa) |
|---|---|---|---|---|---|---|
| PF | 12 | 0.04 | 1.3 | 7 | 1600 | 42 |

*2.2. Mix Proportions*

According to our team's existing research results [12,17], as well as some previous studies [19–21], the IOT replacement ratio was 0%, 20%, 40%, 60%, 80%, and 100%, respectively. The water/cementitious and cementitious/aggregates were set at 0.3 and 0.45, respectively. The volume content of PF was designed to be 2% [3,18]. Table 4 lists the mix proportions for the six groups.

**Table 4.** Mix proportions (kg m$^{-3}$).

| Type | S | IOTs | Water | Fly Ash | Metakaolin | Cement | Superplasticizer | PVA |
|---|---|---|---|---|---|---|---|---|
| IOT0-P2.0 | 565.6 | 0 | 440.3 | 377.4 | 188.7 | 692 | 5.66 | 26 |
| IOT20-P2.0 | 452.48 | 113.12 | 440.3 | 377.4 | 188.7 | 692 | 5.66 | 26 |
| IOT40-P2.0 | 339.36 | 226.24 | 440.3 | 377.4 | 188.7 | 692 | 5.66 | 26 |
| IOT60-P2.0 | 226.24 | 339.36 | 440.3 | 377.4 | 188.7 | 692 | 5.66 | 26 |
| IOT80-P2.0 | 113.12 | 452.48 | 440.3 | 377.4 | 188.7 | 692 | 5.66 | 26 |
| IOT100-P2.0 | 0 | 565.6 | 440.3 | 377.4 | 188.7 | 692 | 5.66 | 26 |

In the nomenclature, "IOT20-P2.0" is used as an example, where "IOT" refers to iron ore tailing. "P" stands for polyvinyl alcohol (PVA) fiber. The numbers "20" and "2.0" represent the IOT replacement ratio of 20% and PF volume content of 2.0%, respectively.

*2.3. Experimental Instruments*

The testing instrument mainly comprised a computer-controlled electro-hydraulic servo universal testing machine from MTS, an NMR instrument (MacroMR12-150H-I) manufactured by New'mai Company (Suzhou, China), and a rapid freeze–thaw (F-T) machine. In addition, S-4800 cold field emission SEM was also used.

*2.4. Research Program*

The research program for this paper is presented in Figure 3. Six groups of mixtures were prepared and then studied. The mechanical properties, sulfate F-T cycles, pore structure, as well as microscopic characteristics of the IOT-ECC were evaluated.

*2.5. Test Method*

According to the Chinese national standard GB/T 50081-2018 and CESC13-2009, the $f_{cu}$, four-point $f_f$, and $f_c$ were evaluated using 100 mm × 100 mm × 100 mm cubes, 300 mm (length) × 76 mm (width) × 13 mm (thickness) plates, and Φ100 mm (diameter) × 200 mm (high) cylinders, respectively. In addition, the control method of displacement was applied for $f_c$ test and the loading speed selected was 0.1 mm/min.

The sulfate F-T resistance test was conducted using the rapid F-T machine (temperature of the specimen center from −18.0 ±1 to 5.0 ±1 °C) following the Chinese standard GB/T 50082-2009. A solution of Na$_2$SO$_4$ with a concentration of 5% was utilized. Before the test, the specimens—that is, (100 × 100 × 100 mm) cubes and (100 × 100 × 400 mm) prisms—were placed in the water and immersed for 4 days and then were dried, to test the initial compressive strength, initial elastic modulus, and initial mass, respectively. Finally, the

specimens were placed in the automatic rapid concrete freeze–thaw machine and related indexes were tested after every 50 F-T cycles.

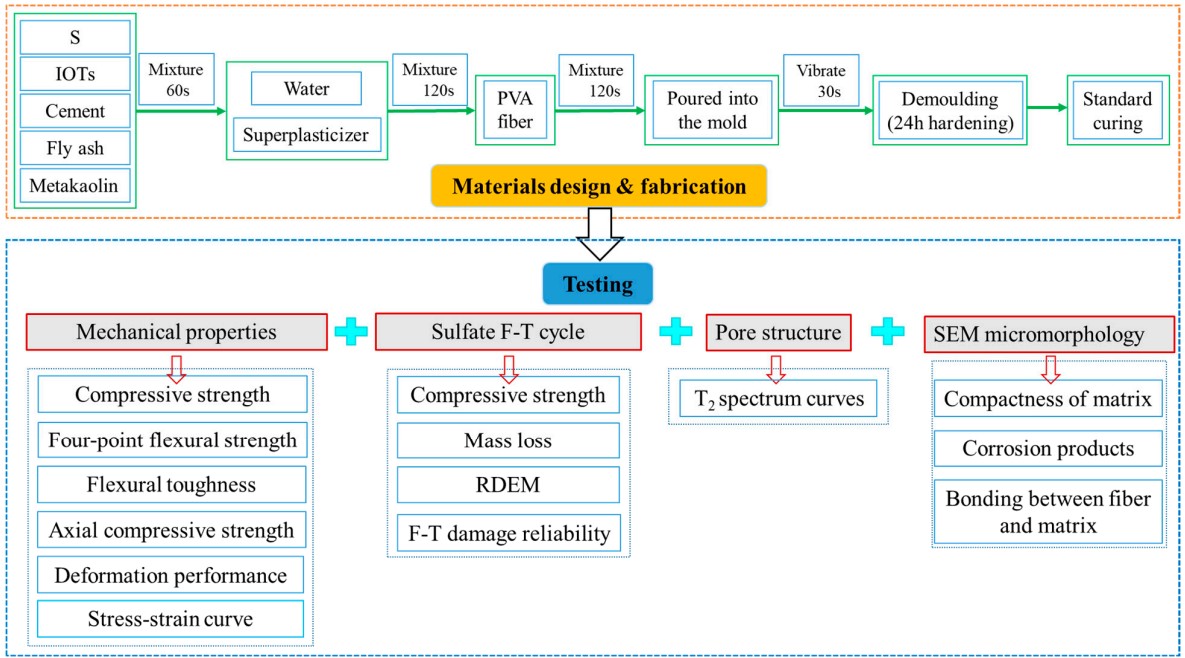

**Figure 3.** Research program of IOT-ECC.

In terms of microstructure, the pore structure and micro-morphology were tested. Before conducting the pore structure test, specimens were saturated in a pressurized device to allow complete water penetration before testing, after which the test was carried out. In addition, the specimens with sizes being approximately 5 mm × 5 mm × 2 mm were taken from 100 mm × 100 mm × 100 mm specimens with varying sulfate F-T cycles and then were imaged using an S-4800 cold field emission SEM. The prepared samples underwent pre-treatment, which involved drying at 40 °C for 24 h and coating with gold spray before testing.

## 3. Mechanical Properties of IOT-ECC

### 3.1. Compressive Strength

The $f_{cu}$ of IOT-ECCs was tested after 28 d, as shown in Figure 4. It can be seen from Figure 4a that the $f_{cu}$ of the IOT-ECCs increases first and then decreases as the IOT replacement ratio increases. Upon reaching 80% of the IOT replacement ratio, the IOTs have a positive effect on the $f_{cu}$ of the IOT-ECCs, especially 40%. For example, the $f_{cu}$ of IOT20-P2.0, IOT40-P2.0, IOT60-P2.0, and IOT80-P2.0 are increased by 10.73%, 15.94%, 7%, and 8.7%, respectively, compared with IOT0-P2.0. Similar observations have been observed in previous studies [22]. However, under the 100% IOT replacement ratio, the $f_{cu}$ of IOT100-P2.0 is lower than that of IOT0-P2.0. Similarly, Zhang et al. [23] and Zhao et al. [5] reported that the $f_{cu}$ of ultra-high-performance concrete with a 100% IOT replacement ratio decreased by 9.5% and 14%. This is because internal pores could be filled and optimized by the fine IOT particles when incorporating up to 80% IOTs, increasing the bonding stress between the PF and the matrix [24], resulting in the spatial grid system formed by the randomly distributed fibers in the matrix being firm [25,26]. However, when incorporating a 100% IOT replacement ratio, the decrease in cement slurry per unit area is caused by the high specific surface area of the IOTs, which reduces the bonding strength. Another main reason for the decrease in $f_{cu}$ is due to the formation of hydration products around the un-hydrated cement, which hinders further hydration, resulting in increased porosity. This is also confirmed in Sections 5 and 6.

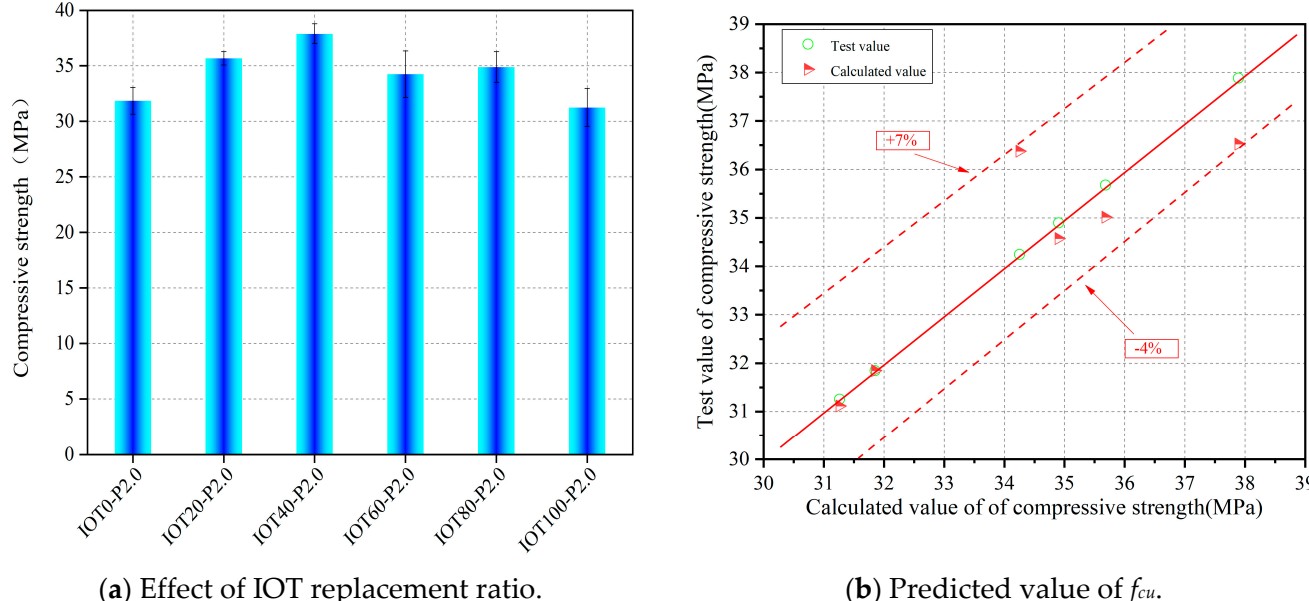

(**a**) Effect of IOT replacement ratio.  (**b**) Predicted value of $f_{cu}$.

**Figure 4.** IOT-ECC of $f_{cu}$ at 28 d.

The relationship between the IOT replacement ratio and the $f_{cu}$ obtained is shown in Equation (1). Figure 3 shows the comparison between the experimental values and the predicted values calculated by Equation (1). It can be seen that the error range is $-4$–7%, which is relatively small.

$$f_{cu,i} = f_{cu,0}(1 + 0.00627i - 0.000065i^2) \tag{1}$$

where $i$ is the IOT replacement ratio. The $f_{cu,0}$ presents the compressive strength of ECCs without IOTs. The $f_{cu,i}$ present the compressive strength of ECCs with IOTs.

### 3.2. Four-Point Flexural Strength

The $f_f$ test results are shown in Figure 5. Similar to the changing trend of the $f_{cu}$, an increase initially and then a decreasing trend is also observed in the $f_f$ as the IOT replacement ratio increases. The $f_f$ of IOT40-P2.0 reaches the maximum value at the 40% IOT replacement ratio. Unfortunately, when the IOT replacement ratio is more than 80%, the $f_f$ of the IOT-ECC is reduced. For example, the $f_f$ of IOT20-P2.0, IOT40-P2.0, and IOT60-P2.0 increased by 8.17%, 14%, and 10.3%, respectively. The $f_f$ of IOT80-P2.0 and IOT100-P2.0 decreased by $-1.7$% and $-3.14$%, respectively, compared with IOT0-P2.0. This is mainly due to the fact that the fine particles of IOTs can effectively fill the pores inside the matrix and at the interface of the PF and the matrix so that the crack resistance of the IOT-ECC is further strengthened by the increased bonding stress between the matrix and the PF. However, when the IOT replacement ratio is large, the amount of cement slurry attached to the IOTs per unit area is reduced, which weakens the bonding matrix and the PF. Meanwhile, poor gradation increases in the porosity result in the deterioration of the bonding between the matrix and the PF. Macroscopically, the $f_f$ is reduced.

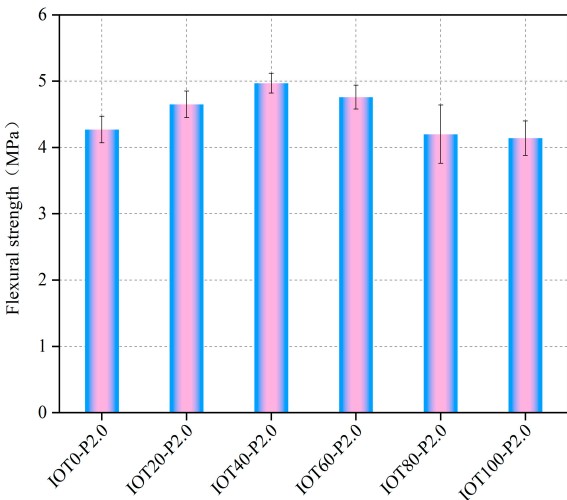

**Figure 5.** IOT-ECC of $f_f$ at 28d.

It is well known that Equation (2) was used to describe the correlation between the $f_{cu}$ and $f_f$. Various parameters $a$ and $b$ have been proposed by several national standards. For example, ACI(318-95) [27]: $a = 0.54$, $b = 0.5$; CEB-FIP: $a = 0.81$, $b = 0.5$. The relationship between the $f_{cu}$ and $f_f$ is fitted, as shown in Equation (3). The comparison between the test results and calculation results is shown in Table 5. The results obtained using the ACI(318-95) standard are relatively lower, with an error range from $-24\%$ to 33.6%, while the calculation results of CEB-FIP are higher and the error range is 0.1–13.9%. In summary, the results calculated by the ACI (318-95) and CEB-FIP are higher than those of the experimental results, making it difficult to apply them directly. However, the difference between the experimental results and the calculated values using Equation (3) is relatively small, with an error range from $-7.7\%$ to 5.5%.

$$f_f = a \times f_{cu}{}^b \tag{2}$$

$$f_f = 0.75 f_{cu}{}^{0.5} \tag{3}$$

**Table 5.** Comparison results between calculated and test.

|  | **Equation (3)** | **ACI (318-95)** | **CEB-FIP** |
|---|---|---|---|
| Error range | $-7.7$–5.5% | $-24$–$-33.6\%$ | 0.1–13.9% |

*3.3. Evaluation of Flexural Toughness*

3.3.1. Load–Deflection Curve

The load–deflection curves under the four-point flexural test of the IOT-ECCs are shown in Figure 6. Each load–deflection curve shows obvious deflection hardening. When the IOT replacement ratio is up to 80%, the load–deflection curve is full and the deflection-hardening part is obvious, especially for IOT40-P2.0. As shown in Figure 7, the specimens of IOT40-P2.0 and IOT60-P2.0 are broken by multiple cracks, while the number of cracks in IOT100-P2.0 is less.

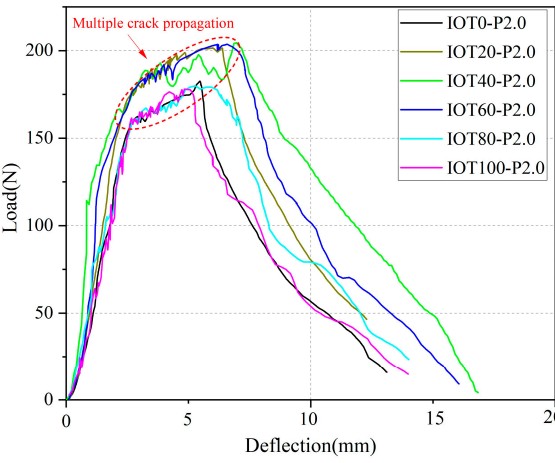

**Figure 6.** Load–deflection curve.

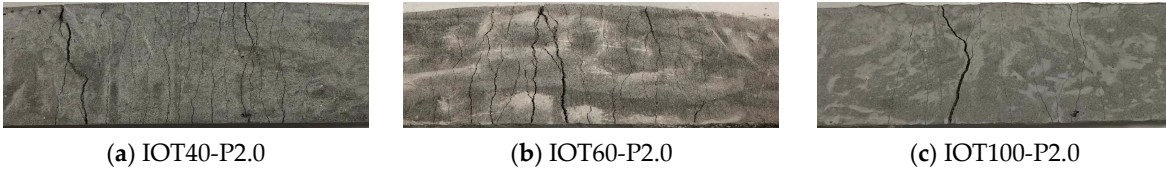

(**a**) IOT40-P2.0　　　　　　　　　　(**b**) IOT60-P2.0　　　　　　　　　　(**c**) IOT100-P2.0

**Figure 7.** Typical multi-crack phenomenon.

### 3.3.2. Flexural Toughness Index

Flexural toughness is also an important indicator of material performance. At present, the flexural toughness of composite materials can be evaluated according to standards such as JSCE-SF4 in Japan, ASTM C1018 in the United States, CECS 13-2009 in China, and RILEM TDF-C162 in Europe, which define and evaluate flexural toughness from different angles. Among many evaluation methods, the ASTMC 1018 standard, as the basic evaluation method, has been widely used because of its clear physical meaning and independence from the size and shape of specimens. Therefore, the ASTMC 1018 standard was used to calculate the toughness index, with the results shown in Table 6.

**Table 6.** Toughness indexes.

| Type | $I_5$ | $I_{10}$ | $I_{20}$ |
|---|---|---|---|
| IOT0-P2.0 | 5.63 | 10.35 | 15.67 |
| IOT20-P2.0 | 6.45 | 14.95 | 25.89 |
| IOT40-P2.0 | 6.98 | 15.26 | 27.70 |
| IOT60-P2.0 | 6.57 | 14.22 | 26.95 |
| IOT80-P2.0 | 5.98 | 10.82 | 17.95 |
| IOT100-P2.0 | 5.3 | 10.44 | 12.95 |

The flexural toughness indexes $I_5$, $I_{10,}$ and $I_{20}$ are calculated using Equations (4)–(6), respectively.

$$I_5 = \frac{\Omega_{3.0\delta}}{\Omega_\delta} \tag{4}$$

$$I_{10} = \frac{\Omega_{5.5\delta}}{\Omega_\delta} \tag{5}$$

$$I_{20} = \frac{\Omega_{10.5\delta}}{\Omega_\delta} \tag{6}$$

where δ is the mid-span deflection corresponding to the initial cracking point of the specimen (mm); $\Omega_\delta$, $\Omega_{3.0\delta}$, $\Omega_{5.5\delta}$, and $\Omega_{10.5\delta}$ are the areas under the load–deflection curves (N.mm) corresponding to the mid-span deflections of δ, 3.0δ, 5.5δ, and 10.5δ, respectively.

The toughness index of the IOT-ECCs was calculated. As the IOT replacement ratio increases, the toughness indexes (including $I_5$, $I_{10,}$ and $I_{20}$) of the IOT-ECCs first increase and then decrease, as shown in Table 6. The toughness indexes of IOT40-P2.0 are higher than those of others. For example, the $I_{20}$ of IOT40-P2.0 increases by 43.7%, compared with IOT0-P2.0. The primary factor is that the "filling effect" of IOTs becomes prominent at an IOT replacement ratio of 40%, which reflects a significantly stronger bonding stress between the PF and the matrix.

### 3.4. Axial Compressive Strength

The $f_c$ is also a crucial parameter for evaluating the mechanical properties of IOT-ECCs in engineering design. Figure 8 illustrates in detail the effect of the IOT replacement ratio on the $f_c$. Similarly, the $f_c$ of IOT40-P2.0 with a 40% IOT replacement ratio undergoes a marked increase, which increases by 12.5%. In contrast, the $f_c$ of IOT100-P2.0 shows a reduction of 5.7% when IOTs are completely replaced. Figure 9a–d show the failure phenomena. No spalling occurred in the samples as the IOT replacement ratio increased.

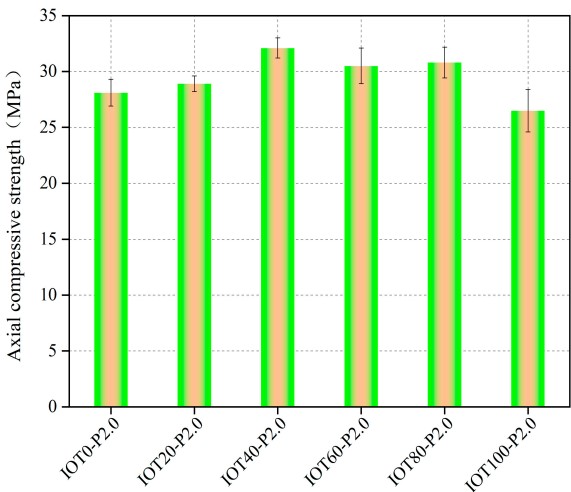

**Figure 8.** $f_c$ of IOT-ECCs at 28 d.

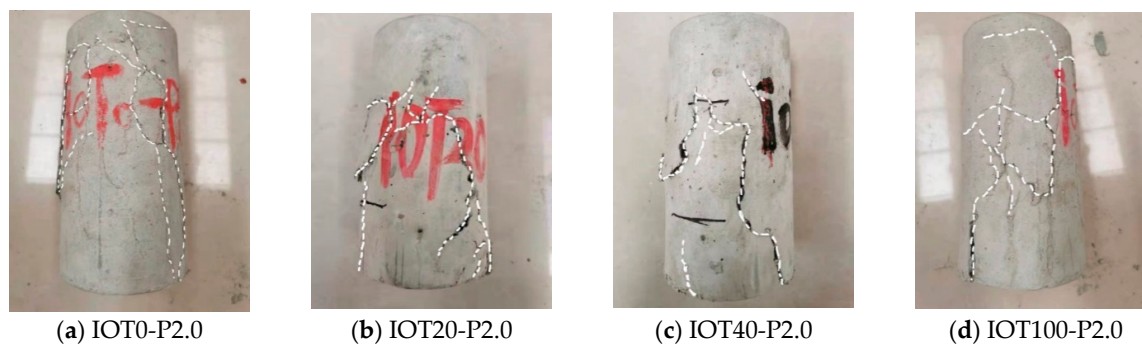

| (**a**) IOT0-P2.0 | (**b**) IOT20-P2.0 | (**c**) IOT40-P2.0 | (**d**) IOT100-P2.0 |

**Figure 9.** Failure phenomena under axial compression.

Similar to the $f_f$ discussed above, the $f_c$ could also be expressed by $f_{cu}$. A linear relationship is established, as shown in Equation (7).

$$f_c = 0.86 f_{cu} \tag{7}$$

*3.5. Deformation Performance*

The deformation performance of the IOT-ECCs can be evaluated by peak strain and elastic modulus, as shown in Table 7. As the IOT replacement ratio increases, the increase in elastic modulus is not significant, while the peak strain of the IOT-ECCs first increases and then decreases. Under a 40% IOT replacement ratio, the peak strain of IOT40-P2.0 increases by 9.7%.

**Table 7.** Test results of the peak strain and elastic modulus.

| Type | $f_c$ (MPa) | Peak Strain ($\times 10^{-3}$) | Elastic Modulus (GPa) |
|---|---|---|---|
| IOT0-P2.0 | 28.1 | 3.92 | 22.4 |
| IOT20-P2.0 | 28.9 | 4.16 | 22.3 |
| IOT40-P2.0 | 32.1 | 4.34 | 22.9 |
| IOT60-P2.0 | 30.5 | 3.96 | 22.7 |
| IOT80-P2.0 | 30.8 | 3.89 | 22.2 |
| IOT100-P2.0 | 26.5 | 3.91 | 21.9 |

*3.6. Compressive Stress–Strain Curve*

The stress–strain curves of the IOT-ECCs under axial compression are shown in Figure 10. The ascending parts of the curves are approximately the same, and stress increases linearly before reaching peak stress. Beyond the point of maximum stress, the capacity to bear loads reduces, while the descending segments of the curves become steeper with the IOT replacement ratio.

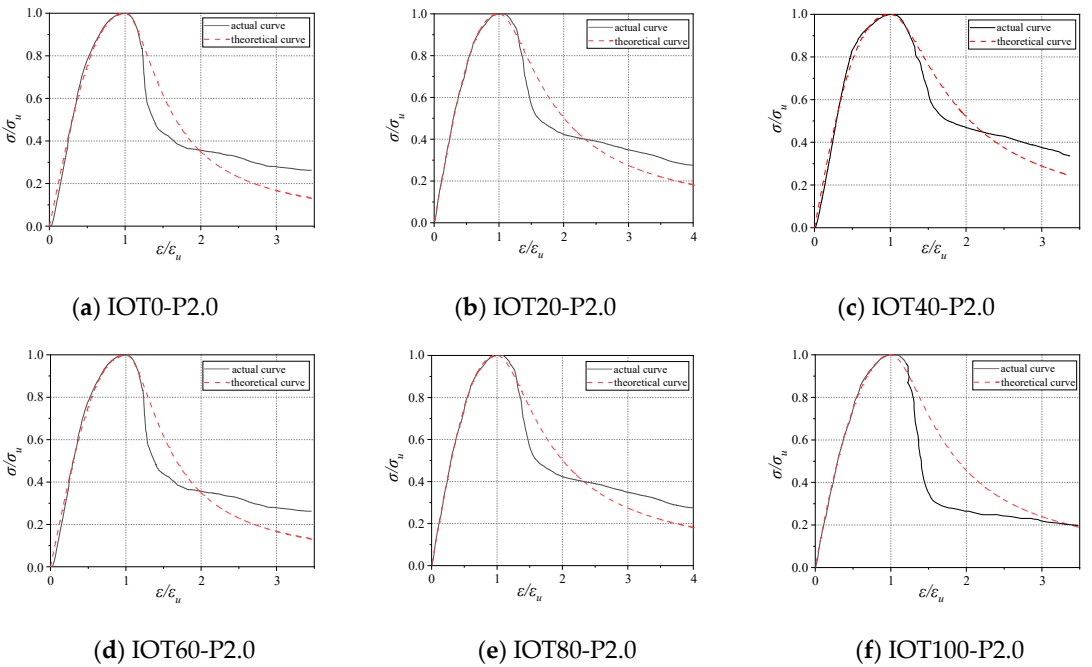

(**a**) IOT0-P2.0    (**b**) IOT20-P2.0    (**c**) IOT40-P2.0

(**d**) IOT60-P2.0    (**e**) IOT80-P2.0    (**f**) IOT100-P2.0

**Figure 10.** Axial compressive stress–strain curve.

A piecewise expression can be used to represent the dimensionless stress–strain curve, as described in the literature [28], as shown in Equations (8) and (9):

$$y = \frac{Ax}{1 + (2A - 3)x^2 + (2 - A)x^3}, 0 \leq x \leq 1 \tag{8}$$

$$y = \frac{x}{B(x - 1)^2 + x}, x \geq 1 \tag{9}$$

where $x = \varepsilon/\varepsilon_{cr}$, $y = \sigma/\sigma_{cr}$; $\varepsilon_{cr}$ represents peak strain; $\sigma_{cr}$ represents peak stress; and $A$ and $B$ represent the parameters of the shape of the curve.

Nonlinear regression analysis was carried out on the dimensionless stress–strain curve using Equations (8) and (9), and the theoretical curve obtained is shown in Figure 10. Table 8 displays the values of parameters A and B, and the correlation coefficients ($R^2$). It can be found that the theoretical curve obtained is in excellent agreement with the test curve.

**Table 8.** Curve-fitting parameters of stress–strain.

| Type | Ascending Section | | Descending Section | |
|---|---|---|---|---|
| | *A* | $R^2$ | *B* | $R^2$ |
| IOT0-P2.0 | 1.749 | 0.99645 | 1.847 | 0.8352 |
| IOT20-P2.0 | 1.733 | 0.99949 | 1.985 | 0.80681 |
| IOT40-P2.0 | 1.804 | 0.9906 | 2.380 | 0.8374 |
| IOT60-P2.0 | 1.737 | 0.99278 | 2.024 | 0.88052 |
| IOT80-P2.0 | 1.725 | 0.9985 | 1.913 | 0.8569 |
| IOT100-P2.0 | 1.696 | 0.9914 | 2.265 | 0.8488 |

## 4. Sulfate F-T Resistance of IOT-ECCs

### 4.1. Compressive Strength

After 50, 100, 150, and 200 F-T cycles, the $f_{cu}$ of the IOT-ECCs under the sulfate F-T cycle is shown in Figure 11. The $f_{cu}$ of each species quickly decreases with the F-T cycle. For example, the $f_{cu}$ of IOT0-P2.0, IOT20-P2.0, IOT40-P2.0, IOT60-P2.0, IOT80-P2.0, and IOT100-P2.0 after 200 F-T cycles decreased by 54.19%, 46.13%, 42.17%, 42.33%, 52.4%, and 59.56%, respectively. The main cause of this is the gradual increase in internal deformation with the F-T cycle. Up to 80% IOTs can inhibit the decrease in $f_{cu}$, especially 40%, which is significantly higher than that of IOT0-P2.0. However, a 100% IOT replacement ratio accelerates $f_{cu}$ reduction.

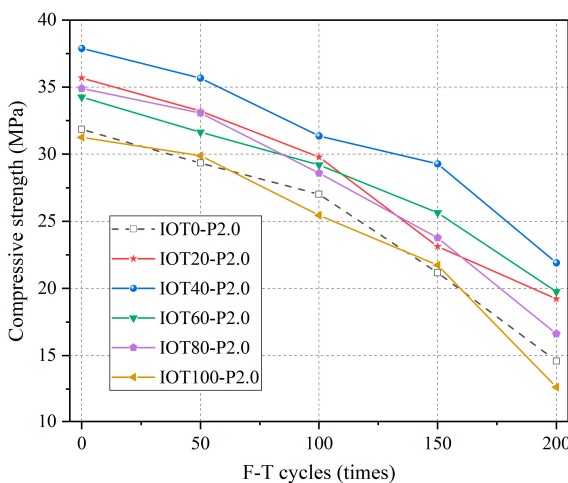

**Figure 11.** $f_{cu}$ of IOT-ECC under sulfate F-T cycle.

### 4.2. Mass Loss

The mass losses of the IOT-ECCs after 50, 100, 150, and 200 F-T cycles are presented in Figure 12. The mass loss of each specimen first increases and then quickly decreases with the F-T cycle. This is because, initially, $SO_4^{2-}$ diffuses by the pores into the specimens, which is consumed by $Ca(OH)_2$, forming $CaSO_4 \cdot 2H_2O$ and $3CaO \cdot Al_2O_3 \cdot CaSO_4 \cdot 31H_2O$, increasing specimen mass, as demonstrated in Equations (10) and (11). As corrosion progresses, an increasing amount of corrosion products (gypsum and ettringite) accumulate, exerting expansion forces on the inner walls of the pores. The pores undergo rapid expansion due to the combined effect of expansion force and F-T force, resulting in the formation

of interconnected micro-cracks that hasten the deterioration of the matrix. After 200 F-T cycles, the mass losses of the IOT-ECCs with 20%, 40% 60%, and 80% IOTs are less than that of the IOT0-P2.0. Unfortunately, the mass loss of the IOT-ECC with 100% increased by 6.3% more than that of the IOT0-P2.0.

$$Na_2SO_4 + Ca(OH)_2 + 2H_2O \rightarrow CaSO_4 \cdot 2H_2O + 2Na_2OH \tag{10}$$

$$4CaO \cdot Al_2O_3 \cdot 12H_2O + 3Na_2SO_4 + 2Ca(OH)_2 + 20H_2O \rightarrow 3CaO \cdot Al_2O_3 \cdot CaSO_4 \cdot 31H_2O + 6NaOH \tag{11}$$

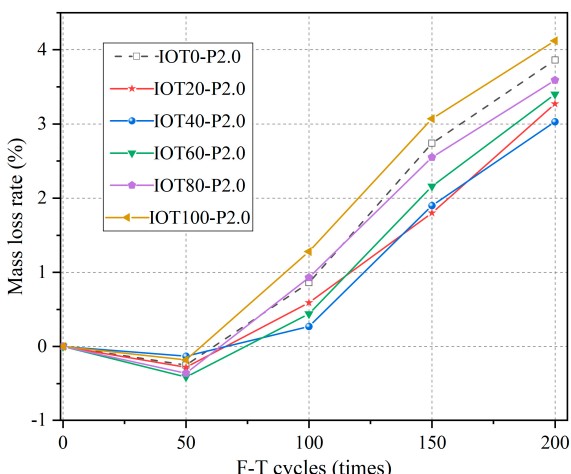

**Figure 12.** Mass loss of IOT-ECCs under sulfate F-T cycle.

*4.3. RDEM*

The RDEM is another crucial indicator for assessing the durability of IOT-ECCs. The RDEM of the IOT-ECCs after 50, 100, 150, and 200 F-T cycles are shown in Figure 13. The RDEM trend follows a similar pattern to the mass loss. The RDEM of the IOT-ECCs increases as a result of effective pore filling by gypsum and ettringite under fewer than 50 F-T cycles. However, the RDEM of each specimen decreases rapidly after 100–200 F-T cycles, especially for IOT100-P2.0. This is mainly because the change in the RDEM is closely linked to the internal pores. The increase in F-T cycles leads to the formation of numerous micro-cracks, which accelerate deterioration and decrease the RDEM. The RDEM of the IOT-ECC with 100% IOTs decreased by 8% more than that of the IOT0-P2.0 after 200 F-T cycles. In contrast, the RDEM of the remaining specimens, including IOT-ECCs with 20%, 40%, 60%, and 80% IOTs, are higher than that of the IOT0-P2.0.

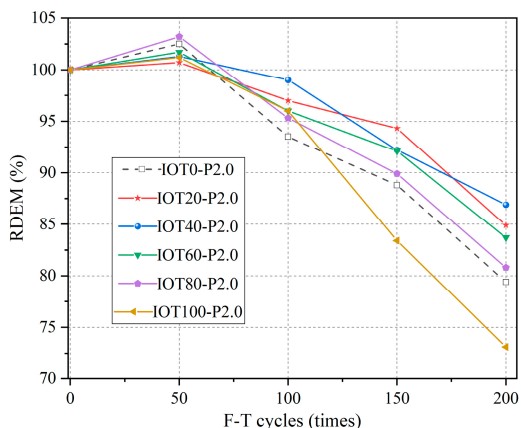

**Figure 13.** RDEM of IOT-ECC under sulfate F-T cycle.

### 4.4. F-T Damage Reliability

Gradual concrete structure failure under the influence of F-T cycles is seen as a process of gradual accumulation of fatigue damage. The performance degradation caused by fatigue damage can be used to analyze the reliability of concrete for a certain period, to comprehensively understand the frost resistance of concrete and accurately quantify the F-T damage. Therefore, referring to the literature [29], the reliability coefficient ($\beta(n)$) of the IOT-ECCs under the sulfate F-T cycle is calculated by Equations (12) and (13) and shown in Figure 14.

$$k_E = \frac{e^{\frac{\sigma_N^2}{2}}}{N_f}; k_D = \frac{e^{\frac{\sigma_N^2}{2}}(e^{\frac{\sigma_N^2}{2}} - 1)}{N_f} \tag{12}$$

$$\beta(n) = \frac{\mu_D}{\sigma_D} = -\frac{\frac{1}{2}\ln(1 + \frac{k_D}{k_E^2 n}) + \ln(k_E n)}{\sqrt{\ln(1 + \frac{k_D}{k_E^2 n})}} \tag{13}$$

where $\sigma_N = \sigma_D$ is the log standard deviation; the $N_f$ is the F-T damage life; and the $k_E$ and $k_D$ are expectation and variance. The $n$ is the time of F-T cycles. The $\mu_D$ is the logarithmic average of instantaneous cumulative damage and the $\beta(n)$ is the reliability coefficient.

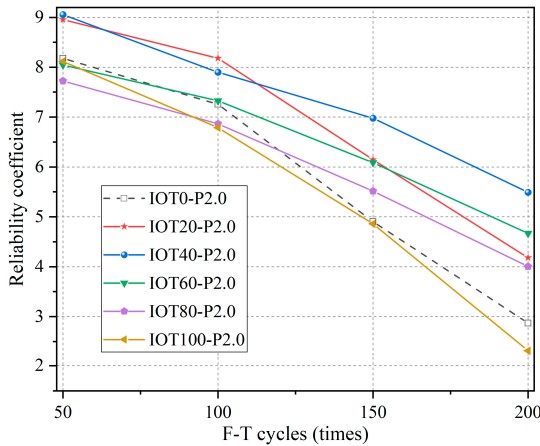

**Figure 14.** $\beta(n)$ of IOT-ECCs under sulfate F-T cycle.

The calculation results of the F-T damage reliability analysis model are shown in Figure 14. The $\beta(n)$ decreases as the number of F-T cycles increases. The $\beta(n)$ of IOT0-P2.0, IOT20-P2.0, IOT40-P 2.0, and IOT60-P2.0 remain at a high level after 200 F-T cycles, especially IOT40-P2.0. The $\beta(n)$ of IOT100-P2.0 is lower than that of IOT0-P2.0. The $\beta(n)$ of the IOT-ECCs from highest to lowest is IOT40-P2.0 > IOT60-P2.0 > IOT20-P2.0 > IOT80-P2.0 > IOT100-P2.0, which is consistent with the test results. This indicates that the reliability model can better reflect the F-T damage to the IOT-ECCs.

## 5. Pore Structure of IOT-ECCs

Figure 15 shows the $T_2$ spectrum curves of the IOT-ECCs. Three signal peaks including a main signal peak and two secondary signal peaks are observed in each $T_2$ spectrum curve, and the primary signal peak has a significantly higher signal intensity than the secondary signal peaks, as shown in Figure 15a,b. This suggests that the specimen contains a greater number of micro-pores, compared to medium and large pores, and the matrix mainly contains micro-pores [30,31]. The $T_2$ spectral curve shows a rightward shift as an increase in both the IOT replacement ratio and F-T cycles, which indicates that the percentage of medium and large pores increases.

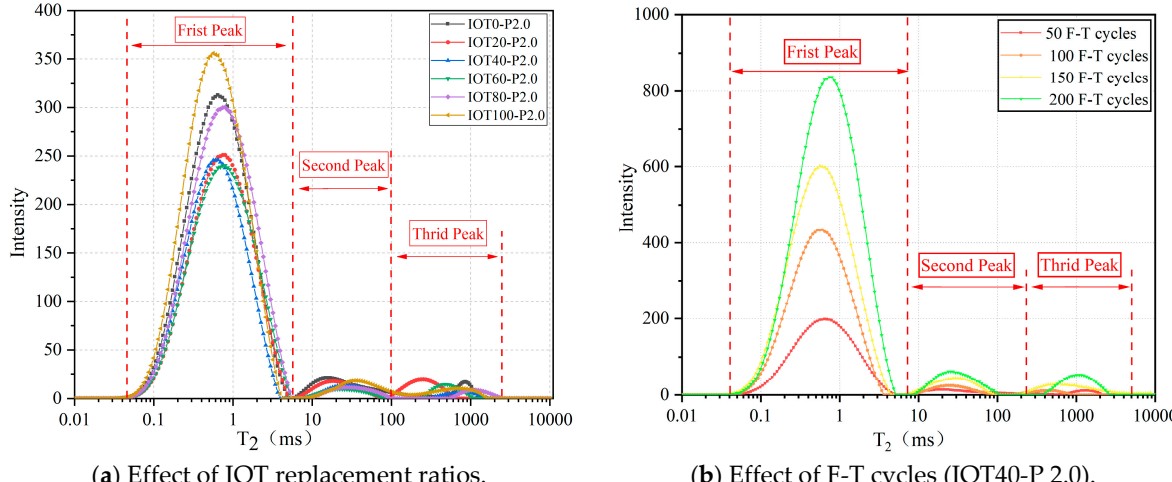

(**a**) Effect of IOT replacement ratios.

(**b**) Effect of F-T cycles (IOT40-P 2.0).

**Figure 15.** $T_2$ spectrum curves of IOT-ECCs.

The area of each $T_2$ spectrum curve's three peaks is presented in Figure 16a,b. When replacing S with 20–80% IOTs, the peak areas are reduced by 15.78%, 33.75%, 18.8%, and 1.19%, respectively, indicating that incorporating IOTs can effectively reduce the internal pores, especially 40%, which is one of the reasons why IOT40-P2.0 has the highest strength compared to that of others. Under a 100% IOT replacement ratio, the internal porosity of IOT100-P2.0 is higher than that of IOT0-P2.0. Moreover, the peak areas increase gradually with the increase in the F-T cycle, especially more than 100 F-T cycles, which increased the peak areas significantly.

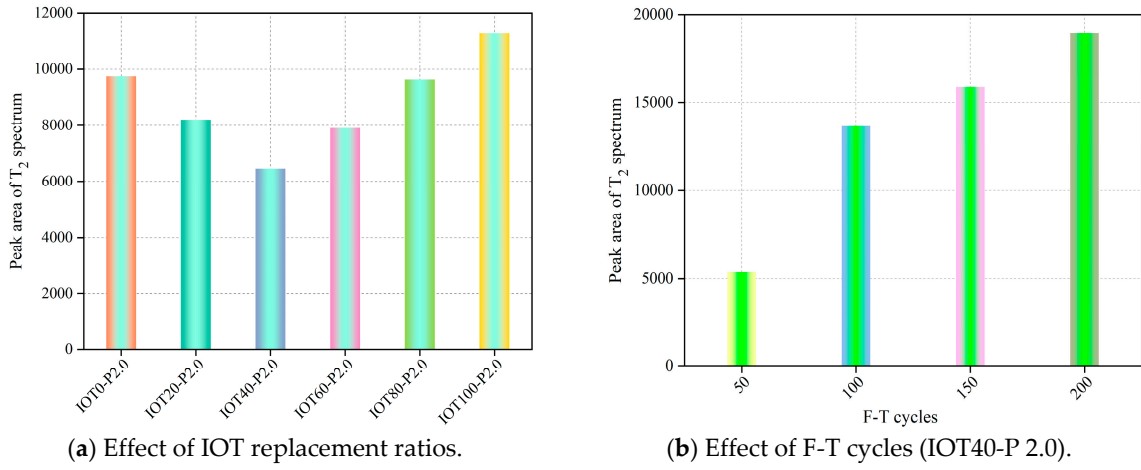

(**a**) Effect of IOT replacement ratios.

(**b**) Effect of F-T cycles (IOT40-P 2.0).

**Figure 16.** Peak areas of $T_2$ spectrum curves of IOT-ECCs.

## 6. SEM Micro-Morphology

The micro-morphology of the IOT-ECCs is shown in Figure 17. As can be seen from Figure 17a–c, the IOT0-P2.0 matrix contains a small amount of Ca $(OH)_2$, cracks, and connected pores. Under a 40% IOT replacement ratio, a significant reduction in the number of connected pores and micro-pores of the IOT40-P2.0 matrix is observed, while Ca $(OH)_2$ is eliminated. In addition, many amorphous C-S-H gels are generated. However, the total number of connected pores and pores in the IOT100-P2.0 matrix has a noticeable increase when S is completely replaced by IOTs. Meanwhile, the density of the matrix is relatively poor compared with IOT40-P2.0 and IOT0-P2.0. Figure 17d–f show that within the pores and cracks of the IOT40-P2.0, corrosion products are formed, including rod-like gypsum and acicular ettringite after different F-T cycles. With the increase in F-T cycles, the dimensions of the pores and cracks are progressively expanding. Meanwhile, the

abundance of acicular ettringite is observed near pores and cracks, especially at 200 F-T cycles. Figure 17g–i found that the PF and the matrix have good adhesion after 50 F-T cycles, and there are few pores at the bonding interface. However, after 100 F-T cycles, there are many pores at the bonding interface of the PF and the matrix. The bonding interface of the PF and the matrix significantly deteriorated after 200 F-T cycles.

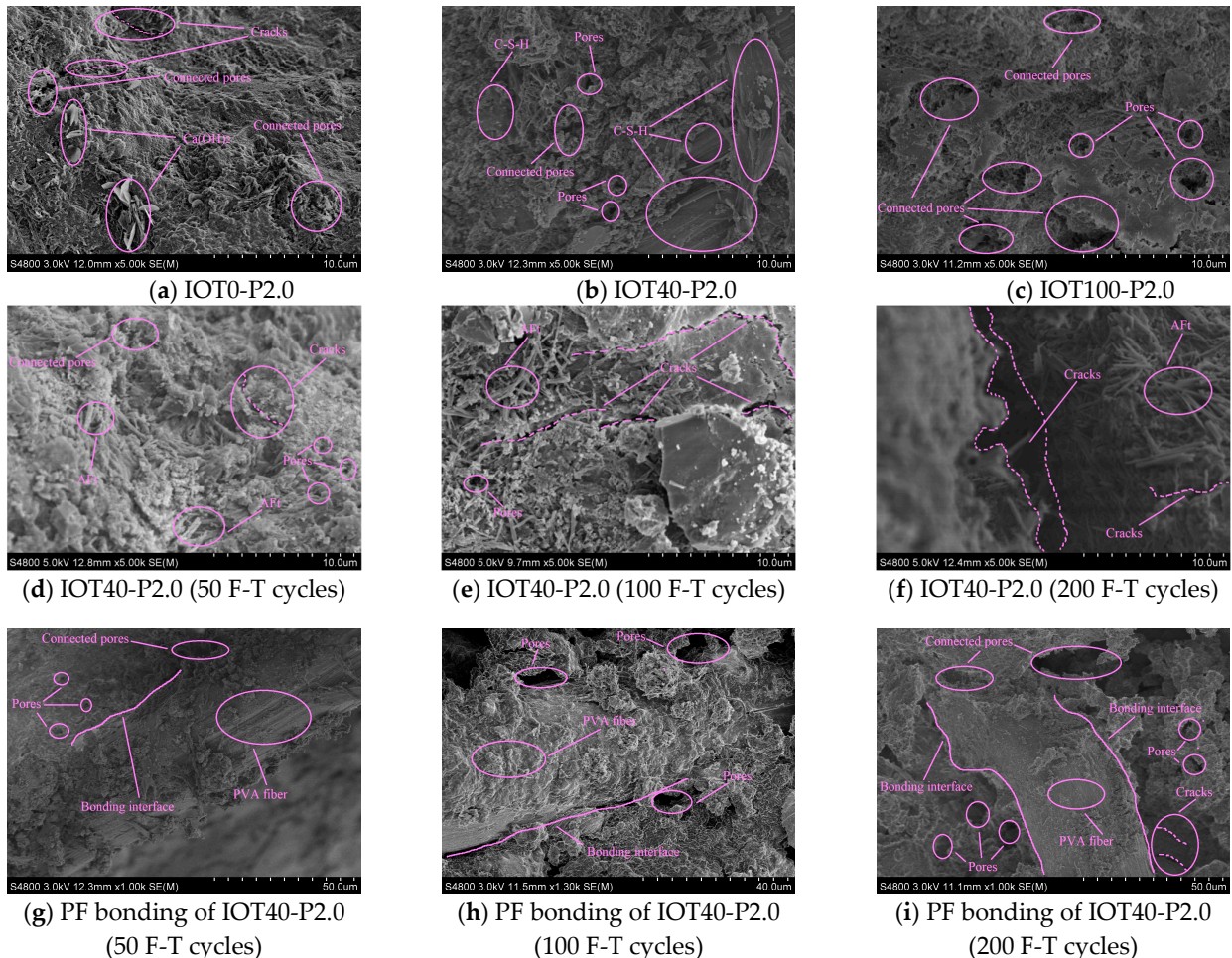

**Figure 17.** SEM micro-morphology.

## 7. Conclusions

This study explored the mechanical properties, sulfate F-T cycle resistance, pore structure, and micro-morphology of IOT-ECCs with different replacement ratios of IOTs. Some conclusions were obtained:

(1) The chemical composition and physical properties of IOTs confirmed the feasibility of preparing IOT-ECCs using IOTs as fine aggregates instead of S. By incorporating up to 80% IOTs, the mechanical properties of IOT-ECCs were improved. IOT-ECCs with 40% IOTs (namely, IOT40-P2.0) can ensure the highest mechanical properties.

(2) As the F-T cycles increased, the mass loss and RDEM of the IOT-ECCs increased first and then rapidly decreased, while the $f_{cu}$ decreased. The F-T damage in the IOT-ECC incorporating 40% IOTs was the smallest, which reflected the best reliability.

(3) NMR results found that when replacing S with 20–80% IOTs, it was beneficial to optimize the internal pore structure of the IOT-ECCs, especially at the 40% IOT replacement ratio. The pore structure of IOT-ECCs gradually deteriorated as the F-T cycle increased, and exhibited a notable deterioration after 100 F-T cycles.

> (4) The SEM results showed that the compactness of IOT-ECCs was improved substantially by incorporating 40% IOTs. After 100 F-T cycles, the dimensions of the pores and cracks and the bonding interface of the PF and the matrix significantly deteriorated.

The macroscopic and microscopic results showed that using 40% IOTs to replace S to produce IOT-ECCs resulted in superior mechanical properties and durability compared to ordinary ECCs. It can be applied to practical projects such as seismic walls, beams, and columns with energy-dissipating joints, and in the reinforcement of masonry structures. The successful application of industrial by-product IOTs as eco-friendly materials will further promote the sustainable development of the construction industry. Further studies on structural members (beams, joints, and walls) should be conducted.

**Author Contributions:** K.L.: conceptualization, methodology, and writing—original draft. S.W.: conceptualization, investigation, and funding acquisition. X.Q.: methodology, resources, investigation, and writing—review and editing. J.W.: resources. J.X.: methodology and formal analysis. N.Z.: investigation. B.L.: data curation and formal analysis. All authors have read and agreed to the published version of the manuscript.

**Funding:** This research was funded by J.X. grant number [No.2021SF-521]. This research was funded by B.L. grant number [No.2022SF-375].

**Data Availability Statement:** The data presented in this study are available on request from the corresponding author.

**Acknowledgments:** This work was supported by the Project on Key Research and Development of Shaanxi, China (No. 2021SF-521 and 2022SF-375).

**Conflicts of Interest:** The authors declare that they have no known competing financial interests or personal relationships that could have appeared to influence the work reported in this paper.

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
