# Peer review of "Development of Engineered Cementitious Composites (ECCs) Incorporating Iron Ore Tailings as Eco-Friendly Aggregates"

_buildings, doi:10.3390/buildings13051341_

Round 1

Reviewer 1 Report

Please read the attachment. Thank you. 

Reviewer 2 Report

Authors done an excellent work on the "Mechanical properties and sulfate freeze-thaw resistance of green engineered cementitious composites (ECC) using iron ore tailings industrial by-products as aggregate". To improve the quality the article, comments listed below must be incorporated in the revisions. 

1. The title of the paper is too long. In general, it is not recommended to exceed 10 words.

2. The paper has several typos. Authors need to proofread the paper to eliminate all of them.

3. The text of some figure(s) is too small. Authors should make sure that the text can be read if printed on paper.

4. Some figure(s) are blur. Authors should either use a higher resolution figure(s) or redo them as vector graphics.

5. The reference format is inconsistent. Please check the format carefully and ensure it is consistent for all references.

6. The introduction should clearly explain the key limitations of prior work that are relevant to this paper.

7. The experiments should be updated to include some comparison with newer studies.

8. There is not enough discussion of the experimental results.

9. Some text must be added to discuss the future work or research opportunities

10. Can you explain how the replacement ratio for IOTs was determined

11. Was there any specific reason for choosing the replacement ratios in intervals of 20%

Some sentences are too long. Generally, it is better to write short sentences with one idea per sentence.

Reviewer 3 Report

The paper presents a very useful study about using iron ore tailings as aggregates for cementitious materials.

The entire paper is well written, having a logical structure, with many details and explanations regarding the results and the phenomena occurred.

One drawback is the extensive use of abbreviations, even where these are not really necessary, mainly in text, but, also, sometimes in tables, e.g. "silica sand" = "S", FA, F-T, PF, MTS, NMR, etc... along with truly necessary abbreviations like IOTs, and those well known like SEM or those standing for standards like ACI, they make the reading and understanding more difficult; I would recommend giving up some of them and write the words instead.

Then, the study focuses on a very specific cementitious composite which includes also fly ash and fibres, making it hard to verify and reproduce and limiting its real use.

The introductory part should be revised presenting the necessity of this study and its novelty compared to the previous studies. Also, the references should be detailed, not only mentioned in bulk (9-13 or 14-18); a good example are references 28, 29, 30...

Chapter 2 should be "Materials and methods".  It is well described, although some explanations regarding the use of fly ash, specific fibers (PVC) and MK could help.

Chapter 3 should be "Results and discussions", grouping all the results, mechanical and physical and chemical. Again, the descriptions and explanations regarding the obtained results and their causes are comprehensive. It shows a complete set of tests covering all interesting aspects of the researched material. Figures, most of the equations and tables are self-explanatory. Some corrections could be made:

- captions of fig. 3 and 4 

- fig. 3 is not showing the fcu tested after 28d (line 98)

- equation 1 should be explained or referenced 

The conclusions are inline with the results and the present literature, but they should be improved by adding future research and developing directions, possible practical applications, etc.

The references list is adequate.

Therea are minor grammar and speclling errors: use of capital letter where is not necessary (line 28, 63, etc), "um" - table 3, use of past tense intead of present tense (line 54, etc).

Reviewer 4 Report

Reviewer Comments:

The author presents a well-written and well-structured paper, the arguments are clearly developed and the conclusions are soundly-based. The authors presented an experimental study using iron ore tailings, as aggregate, to prepare iron ore tailings-engineered cementitious composites in order to achieve clean production. This is an interesting paper but some modifications must be carried out in order to improve the global quality of the paper, namely:

1.     Please avoid the use of a large number of citations in the same phrase, as “[9-13]” or [14-18]”. It is convenient to explain the major finds of each citation.

2.     Please add a sub-section “Research significance”, in the Introduction section, with the objectives, novelty and findings of this work.

3.     The references should be formatted in accordance with the Journal Guidelines.

4.     Section 2 presents lots of sub-sections, much of them with a few words. Please aggregate.

5.     Please explain the sentence “the experimental results show significant discrepancies from the values calculated by existing standards, making it difficult to apply them directly”

6.     Please present a subsection with a statistical analysis of the experimental results obtained in this work.

7.     Please analyse and compare, with more detail, our experimental results with the results presented in the literature.

8.     Please discuss the vantages and advantages of the use of iron ore tailings as aggregate in comparison with other aggregates.

9.     Please improve the Conclusions section in order to be more concise and direct to the findings and objectives of this work.

I believe the work is suitable for publication in Buildings journal, after major revisions.

No comments.

Round 2

Reviewer 1 Report

Dear Editor and Authors:

 Thank you for providing the point-to-point response.

The authors have carefully and patiently corrected and answered the comments and questions. The manuscript looks perfect now, and the reviewer strongly suggests it be accepted for publication in this journal.

Please feel free to contact me if you have further requests or concerns.

Thank you for reading.

it is fine. just minor errors in grammar and structure.

Reviewer 4 Report

The authors improved the paper in accordance with reviewer suggestion.